Molecular phylogenetics and evolutionary history of the endemic land snail genus Everettia in northern Borneo

Liew Thor-Seng 1 thorsengliew@gmail.com
Marzuki Mohammad Effendi 2
http://orcid.org/0000-0001-6229-0347 Schilthuizen Menno 1 3 4
Chen Yansen 5
Vermeulen Jaap J. 3 6
http://orcid.org/0000-0001-5513-6237 Mohd-Azlan Jayasilan 2
1 Institute for Tropical Biology and Conservation, Universiti Malaysia Sabah , Kota Kinabalu, Sabah , Malaysia
2 Institute of Biodiversity and Environmental Conservation, Universiti Malaysia Sarawak , Kota Samarahan, Sarawak , Malaysia
3 Naturalis Biodiversity Center , Leiden , The Netherlands
4 Institute Biology Leiden, Leiden Univerisity , Leiden , The Netherlands
5 Independent Researcher , Medan, North Sumatra , Indonesia
6 JK Art and Science , Leiden , The Netherlands
Poyarkov Nikolay
Electronic publication date: 2020 Jul 9
Publication date: 2020
Volume: 8
Electronic Location ID: e9416
Received 2019 Oct 30; Accepted 2020 Jun 3
Copyright: © 2020 Liew et al.
Copyright year: 2020
Copyright holder: Liew et al.
License: This is an open access article distributed under the terms of the Creative Commons Attribution License, which permits unrestricted use, distribution, reproduction and adaptation in any medium and for any purpose provided that it is properly attributed. For attribution, the original author(s), title, publication source (PeerJ) and either DOI or URL of the article must be cited.
License URL: https://creativecommons.org/licenses/by/4.0/

Keywords: Mount Kinabalu, Sabah, Sarawak, Kalimantan, Dyakiidae, Mount Tambuyukon, Biogeography, Species distribution modelling

Funding: Martin Fellowship 2008 from ‘Naturalis’ to Thor-Seng Liew Universiti Malaysia Sabah Research Grant GKP0011-STWN-2016 Universiti Malaysia Sarawak Research Grant GL/F07/UMS/02/2017 This study was supported financially by a Martin Fellowship 2008 from ‘Naturalis’ to Thor-Seng Liew; Universiti Malaysia Sabah Research Grant (GKP0011-STWN-2016); Universiti Malaysia Sarawak Research Grant (GL/F07/UMS/02/2017). There was no additional external funding received for this study. The funders had no role in study design, data collection and analysis, decision to publish, or preparation of the manuscript.

==============================
Borneo has gone through dramatic changes in geology and topography from the early Eocene until the early Pliocene and experienced climatic cycling during the Pleistocene. However, how these changes have shaped the present-day patterns of high diversity and complex distribution are still poorly understood. In this study, we use integrative approaches by estimating phylogenetic relationships, divergence time, and current and past niche suitability for the Bornean endemic land snail genus Everettia to provide additional insight into the evolutionary history of this genus in northern Borneo in the light of the geological vicariance events and climatic fluctuations in the Pleistocene. Our results show that northern Borneo Everettia species belong to two deeply divergent lineages: one contains the species that inhabit high elevation at the central mountain range, while the other contains lowland species. Species diversification in these lineages has taken place before the Pliocene. Climate changes during the Pleistocene did not play a significant role in species diversification but could have shaped contemporary species distribution patterns. Our results also show that the species-rich highland habitats have acted as interglacial refugia for highland species. This study of a relatively sedentary invertebrate supports and enhances the growing understanding of the evolutionary history of Borneo. Species diversification in Everettia is caused by geological vicariance events between the early Miocene and the Pliocene, and the distribution patterns were subsequently determined by climatic fluctuations in the Pleistocene.

Background

Borneo, the third-largest island in the world, is one of the Earth’s biodiversity hotspots (Mackinnon et al., 1996; Myers et al., 2000). Its biodiversity has been shaped by a long history of geological and climatic stability interspersed with periods of upheaval. During the Palaeogene, east and north Borneo was submerged while the rest of Borneo was connected with other parts of Sundaland. Between Eocene and Pliocene, regional tectonic activities have caused the emergence of land and mountain building in Borneo (Hall, 2013), notably: the formation of highlands in central Borneo, the uplifting of Meratus Mountains in southern Borneo, and uplifting of Mount Kinabalu in northern Borneo. The erosion resulting from these mountain-building events have created the land in the northern, eastern and southern parts of Borneo by filling large basins with sediment.

Borneo has been latitudinally stable, and a large part of it has been covered by tropical forest throughout this period (Lumadyo et al., 1993). Widespread evergreen rainforests would have covered much of Sundaland during the early and middle Miocene (De Bruyn et al., 2014). In the Pleistocene, rainforest persisting in some areas of the island were relatively little affected by climatic fluctuations as compared to other parts of Sundaland (Cannon, Morley & Bush, 2009; Wurster et al., 2010; Morley, 2012).

Hence, Borneo is a suitable natural laboratory for tropical evolutionary biology studies. Most of the studies of Borneo taxa have shown that Borneo was already a major evolutionary hotspot and centre of divergence in the pre-Miocene (see review by De Bruyn et al. (2014)) or pre-Pliocene (Nauheimer, Boyce & Renner, 2012; Klaus et al., 2013; De Bruyn et al., 2014; Grismer et al., 2016; Williams et al., 2017; Chua et al., 2017; Chen et al., 2018). In addition, previous studies suggest that contemporary biodiversity richness and distribution patterns have been affected by climatic fluctuations in the Pleistocene (Barkman & Simpson, 2001; Quek et al., 2007; Jalil et al., 2008; Patou et al., 2010; Lim et al., 2010; Lim & Sheldon, 2011; Ueda et al., 2010).

Most of the previous studies use widespread organisms as model taxa to understand how historical processes shaped the genetic and diversity patterns. However, the genetic and diversity patterns of a well-dispersing taxon may be easily diluted and thus impede the interpretations of the events that shape the patterns (Beck & Rüdlinger, 2014; Manthey et al., 2017). Hence taxa that are relatively sedentary and narrowly distributed, and endemic to Borneo are potentially more suitable model organisms. Slow-moving land snails have proven to be excellent model species to understand evolutionary histories at different scales (Davison, 2002; Hugall et al., 2002), which is why we here employ an endemic land snail genus in the context of the historical biogeography of Borneo.

The taxonomy and distribution of the Bornean endemic land snail genus Everettia Godwin-Austen, 1891 has been well documented in northern Borneo (Liew, Schilthuizen & Vermeulen, 2009). This genus is one of the most speciose macro land snails endemic to Borneo. It occupies different habitats from lowland tropical rainforest to highland montane forest, is found in intact forest, degraded forest, swampy forest, coastal forest and islands. A large number of Everettia species are endemic to the highlands of Borneo, and many lowland species show disjunct distribution (Liew, Schilthuizen & Vermeulen, 2009).

In this study, we aimed to provide a temporal framework for the diversification of the different lineages through the reconstruction of a time-calibrated multilocus species tree using relaxed clock models with species distribution modelling during the last glacial maximum (LGM). We examine whether species diversification in Borneo and highland diversity on Mount Kinabalu are either due to Pleistocene climatic fluctuation or earlier Tertiary palaeogeographic events. First, we estimate the phylogeny of Everettia species in Borneo, including species from Sarawak and Kalimantan, by using mitochondrial and nuclear DNA, to elucidate the evolutionary history of Everettia in northern Borneo in the light of the key vicariance events. Second, we construct species distribution models for Everettia species in Sabah, where extensive occurrence data are available, to examine the changes of species distributions during the last glacial period and identify possible refugia during the LGM.

Methods

Taxon sampling

For molecular phylogenetic analysis, we included 71 Everettia specimens representing 16 of the 17 known species from Sabah. Besides, five Everettia species from Kalimantan and four Everettia species from Sarawak were also included (Table 1; Figs. 1 and 2). The specimens were obtained from the following depositories: BORNEENSIS at Universiti Malaysia Sabah, the Sabah Parks Museum (SP), Jaap Jan Vermeulen’s private collection (JJ), Leiden, Naturalis Biodiversity Center, Leiden (RMNH, ZMA), the Natural History Museum, London (BMNH), Mohammad Effendi Marzuki’s private collection (ME) and Yansen Chen’s private collection (YSC). Additional materials were obtained under the permits: Sarawak Forestry: NPW.907.4.4 (Jld.14)-31), WL14/2017; and Sabah Parks: TS/PTD/5/4 Jld.54 (112). For an outgroup taxon, we included two specimens of Quantula striata Gray, 1834, which belongs to the sister genus of Everettia within the family Dyakiidae.

Table 1 Species, voucher specimens, location information, and GenBank accession number.

No.	Species	Voucher specimens	Location	16S	COI	28S	ITS	
1	Quantula striata	BOR/MOL 13939	Singapore	FJ160646	FJ160693	JQ180190	FJ160732	
2	Quantula striata	BOR/MOL 7905	Labuan Island, Sabah, Malaysia	MN564843	MN564863	–	MN596180	
3	Everettia sp. 1	YC collection	Benualawas, Meratus Range, South Kalimantan, Indonesia	MN564844	MN564864	MN619662	MN596181	
4	Everettia sp. 1	YC collection	Benualawas, Meratus Range, South Kalimantan, Indonesia	MN564845	MN564865	MN619663	MN596182	
5	Everettia sp. 1	YC collection	Benualawas, Meratus Range, South Kalimantan, Indonesia	MN564846	MN564866	MN619664	MN596183	
6	Everettia sp. 2	YC collection	Beramba, Meratus Range, South Kalimantan, Indonesia	MN564847	MN564867	MN619665	MN596184	
7	Everettia sp. 2	YC collection	Beramba, Meratus Range, South Kalimantan, Indonesia	MN564848	MN564868	MN619666	MN596185	
8	Everettia sp. 3	YC collection	Desa Tongka, North Barito, Centre Kalimantan, Indonesia	MN564849	MN564869	MN619667	MN596186	
9	Everettia sp. 4	V12508	Sangkulirang, East Kalimantan, Indonesia	–	JQ180089	JQ180188	–	
10	Everettia sp. 5	V12504	Sangkulirang, East Kalimantan, Indonesia	–	JQ180090	JQ180189	–	
11	Everettia sp. 6	BOR/MOL 5480	Lanjak-Entimau Wildlife Sanctuary, Sarawak, Malaysia	JQ180055	JQ180088	–	JQ180114	
12	Everettia sp. 7	BOR/MOL 5481	Lanjak-Entimau Wildlife Sanctuary, Sarawak, Malaysia	JQ180054	JQ180086	JQ180186	JQ180112	
13	Everettia sp. 7	BOR/MOL 5481	Lanjak-Entimau Wildlife Sanctuary, Sarawak, Malaysia	–	JQ180087	–	JQ180113	
14	Everettia baramensis	WM collection	Mulu National Park, Sarawak, Malaysia	JQ180053	JQ180085	JQ180185	JQ180111	
15	Everettia algaia	ME collection	Niah Cave, Miri, Sarawak	–	MN564870	MN619668	MN596187	
16	Everettia corrugata corrugata	BOR/MOL 12936	Mt. Kinabalu northwestern slope, 3,000 m (S142), Sabah, Malaysia	FJ160619	FJ160666	–	FJ160710	
17	Everettia corrugata corrugata	BOR/MOL 12828	Mt. Kinabalu southern slope, 3,400 m (S16), Sabah, Malaysia	FJ160621	FJ160668	JQ180164	FJ160711	
18	Everettia corrugata williamsi	BOR/MOL 12935	Mt. Kinabalu southeastern slope, 3,100 m (S69A), Sabah, Malaysia	FJ160622	FJ160669	JQ180165	FJ160712	
19	Everettia corrugata williamsi	BOR/MOL 12935	Mt. Kinabalu southeastern slope, 3,100 m (S69B), Sabah, Malaysia	JQ180041	JQ180074	JQ180166	JQ180106	
20	Everettia dominiki	BOR/MOL 12861	Mt. Kinabalu southwesthern slope, 2,100 m (S100), Sabah, Malaysia	FJ160598	FJ160649	JQ180180	FJ160696	
21	Everettia dominiki	BOR/MOL 12800	Mt. Tambuyukon eastern slope 2,200 m (S102), Sabah, Malaysia	FJ160599	FJ160650	JQ180181	FJ160697	
22	Everettia dominiki	BOR/MOL 12838	Mt. Kinabalu southeastern slope, 3,100 m (S68), Sabah, Malaysia	FJ160606	FJ160657	JQ180182	FJ160700	
23	Everettia dominiki	BOR/MOL 12860	Mt. Kinabalu southwesthern slope, 3,100 m (S87), Sabah, Malaysia	FJ160607	FJ160658	JQ180183	FJ160701	
24	Everettia planispira	BOR/MOL 14115	Tawau Hills Park, Tawau, Sabah, Malaysia	FJ160595	FJ160647	JQ180177	FJ160694	
25	Everettia monticola	BOR/MOL 12798	Mt. Kinabalu Southern slope, 1,700 m (S32), Sabah, Malaysia	FJ160596	FJ160648	JQ180179	FJ160695	
26	Everettia interior	BOR/MOL 12879	Batu Tinagas, Sapulut, Sabah, Malaysia	FJ160637	FJ160684	–	FJ160725	
27	Everettia interior	BOR/MOL 12871	Batu Sanaron, Sapulut, Sabah, Malaysia	FJ160638	FJ160685	JQ180170	FJ160726	
28	Everettia jasilini	BOR/MOL 12846	Mt. Kinabalu rortheastern slope, 3,100 m (S80), Sabah, Malaysia	FJ160617	FJ160664	JQ180174	FJ160708	
29	Everettia jasilini	BOR/MOL 12810	Mt. Kinabalu rorthwestern slope, 2,800 m (S140), Sabah, Malaysia	FJ160618	FJ160665	JQ180175	FJ160709	
30	Everettia safriei	BOR/MOL 12929	Mt. Kinabalu rortheastern slope, 3,300 m (S79), Sabah, Malaysia	FJ160614	FJ160663	JQ180176	FJ160707	
31	Everettia safriei	BOR/MOL 12855	Mt. Kinabalu southeastern slope, 2,900 m (S66), Sabah, Malaysia	JQ180049	JQ180082	–	JQ180109	
32	Everettia klemmatanica	BOR/MOL 14097	Mt. Kinabalu southern slope, 1,700 m, Sabah, Malaysia	FJ160611	FJ160660	–	FJ160704	
33	Everettia klemmatanica	BOR/MOL	Mahua, Crocker Range, 1,200 m, Sabah, Malaysia	JQ180039	JQ180073	JQ180163	JQ180105	
34	Everettia lapidini	SP 12924	Mt. Kinabalu southwesthern slope, Marai Parai, 1,700 m, (SP12924), Sabah, Malaysia	FJ160645	FJ160692	JQ180168	FJ160731	
35	Everettia layanglayang	BOR/MOL 4578	Mt. Kinabalu northwestern slope, 1,800 m, Sabah, Malaysia	FJ160624	FJ160671	–	FJ160714	
36	Everettia layanglayang	BOR/MOL 4486	Mt. Kinabalu southern slope, 2,300 m (S11), Sabah, Malaysia	FJ160626	FJ160673	–	FJ160716	
37	Everettia layanglayang	SP 12907	Mount Alab, Crocker Range, 1,800 m (SP12907?), Sabah, Malaysia	FJ160644	FJ160691	–	FJ160730	
38	Everettia layanglayang	BOR/MOL 12808	Mt. Kinabalu southern slope, Mesilau, 2,500 m, Sabah, Malaysia	JQ180042	JQ180075	JQ180167	JQ180107	
39	Everettia paulbasintali	BOR/MOL 6399	Tawau Hills Park, Tawau, Sabah, Malaysia	FJ160613	FJ160662	JQ180171	FJ160706	
40	Everettia paulbasintali	BOR/MOL 12821	Tabin Wildlife Reserve (HQ), Lahad Data, Sabah, Malaysia	FJ160642	FJ160689	JQ180172	FJ160729	
41	Everettia paulbasintali	BOR/MOL 13011	Luasing, INIKEA site, Tawau, Sabah, Malaysia	MN564850	MN564871	MN619669	MN596188	
42	Everettia paulbasintali	BOR/MOL 13315	Imbak Crayon Conservation Area, Telupid, Sabah, Malaysia	MN564851	MN564872	MN619670	MN596189	
43	Everettia paulbasintali	BOR/MOL 13320	Imbak Crayon Conservation Area, Telupid, Sabah, Malaysia	MN564852	–	MN619671	MN596190	
44	Everettia paulbasintali	BOR/MOL 13844	Mount Silam, 600 m, Lahad Data, Sabah, Malaysia	–	–	–	MN596191	
45	Everettia subconsul	BOR/MOL 12813	Mt. Tambuyukon eastern slope, 1,100 m (S114), Sabah, Malaysia	FJ160629	FJ160676	–	FJ160719	
46	Everettia subconsul	SP	Ulu Membakut, Crocker Range, Sabah, Malaysia	FJ160630	FJ160677	JQ180154	FJ160720	
47	Everettia subconsul	BOR/MOL	Danum Valley, Lahad Datu, Sabah, Malaysia	FJ160639	FJ160686	–	FJ160727	
48	Everettia subconsul	SP	Nalapak Substesen, Kinabalu Kinabalu Park,, Sabah, Malaysia	FJ160640	FJ160687	–	FJ160728	
49	Everettia subconsul	BOR/MOL 6488	Gaya Island, Kota Kinabalu, Sabah, Malaysia	FJ160634	FJ160681	JQ180155	FJ160722	
50	Everettia subconsul	BOR/MOL 6492	Crocker Range Park, Keningau HQ, 800 m, Sabah, Malaysia	MN564853	MN564873	–	MN596192	
51	Everettia subconsul	BOR/MOL	Danum Valley, Lahad Datu, Sabah, Malaysia	JQ180027	JQ180061	JQ180156	JQ180095	
52	Everettia subconsul	BOR/MOL 13936	Kampung Magnin, Kudat, Sabah, Malaysia	JQ180028	JQ180062	–	JQ180096	
53	Everettia subconsul	BOR/MOL 12868	Kiansom, Crocker Range, Sabah, Malaysia	JQ180029	JQ180063	–	JQ180097	
54	Everettia subconsul	BOR/MOL 12820	Imbak Crayon Conservation Area, Telupid, Sabah, Malaysia	JQ180031	JQ180065	–	JQ180099	
55	Everettia subconsul	SP	Tahubang, Mount Kinabalu, Sabah, Malaysia	–	JQ180066	–	JQ180100	
56	Everettia subconsul	SP	Kinosolopon, Kimanis, Crocker range, Sabah, Malaysia	JQ180033	JQ180068	JQ180157	JQ180102	
57	Everettia subconsul	BOR/MOL 12823	Poring, Mount Kinabalu (600 m), Sabah, Malaysia	JQ180034	JQ180069	–	JQ180103	
58	Everettia subconsul	BOR/MOL 14108	Meliau Range, Sabah, Malaysia	JQ180035	JQ180070	JQ180158	–	
59	Everettia subconsul	BOR/MOL 6485	Lumaku, Sabah, Malaysia	JQ180038	JQ180072	JQ180160	–	
60	Everettia subconsul	BOR/MOL 6783	Sepanggar Island, Sabah, Malaysia	MN564854	MN564874	–	MN596193	
61	Everettia subconsul	BOR/MOL 8852	Gaya Island, Sabah, Malaysia	MN564855	MN564875	MN619672	MN596194	
62	Everettia subconsul	BOR/MOL 8926	Sayap, Mt. Kinabalu, 800 m, Sabah, Malaysia	MN564856	MN564876	–	MN596195	
63	Everettia subconsul	BOR/MOL 9246	Melalap, Crocker Range, 400 m, Sabah, Malaysia	MN564857	MN564877	–	–	
64	Everettia subconsul	BOR/MOL 13018	Inobong, Crocker Range, 300 m, Sabah, Malaysia	MN564858	MN564878	MN619673	MN596196	
65	Everettia themis	SP 12599	TBC Tower, Crocker Range, 1,400 m (SP12599), Sabah, Malaysia	FJ160623	FJ160670	JQ180161	FJ160713	
66	Everettia themis	BOR/MOL	Mt. Kinabalu southern slope, 1,900 m, Sabah, Malaysia	FJ160628	FJ160675	JQ180162	FJ160718	
67	Everettia subconsul	BOR/MOL 13056	Banggi Island, Sabah, Malaysia	MN564859	MN564879	MN619674	MN596197	
68	Everettia subconsul	BOR/MOL 13140	Banggi Island, Sabah, Malaysia	MN564860	MN564880	MN619675	MN596198	
69	Everettia jucunda	BOR/MOL 12870	Klias, Beaufort, Sabah, Malaysia	FJ160635	FJ160682	JQ180153	FJ160723	
70	Everettia jucunda	BOR/MOL	Tiga Island, Sabah, Malaysia	FJ160636	FJ160683	–	FJ160724	
71	Everettia jucunda	BOR/MOL 7916	Labuan Island, Sabah, Malaysia	MN564861	MN564881	MN619676	MN596199	
72	Everettia jucunda	BOR/MOL 8648	Kuraman Island, Sabah, Malaysia	MN564862	MN564882	MN619677	MN596200	
73	Everettia jucundior	BOR/MOL	Tawau Hills Park, Tawau, Sabah, Malaysia	FJ160612	FJ160661	JQ180173	FJ160705	
Note:

Abbreviation for repositories of voucher specimens: BORNEENSIS at Universiti Malaysia Sabah, the Sabah Parks Museum (SP), Jaap Jan Vermeulen’s private collection (JJ), Leiden, Naturalis Biodiversity Center, Leiden (RMNH, ZMA), the Natural History Museum, London (BMNH), Mohammad Effendi Marzuki’s private collection (ME), and Yansen Chen’s private collection (YSC).

Figure 1 The distribution of selected taxa and specimens in Borneo for phylogenetic analysis. The numbers in parentheses refer to specimen numbers of Table 1.

(A) Topography of Borneo and the locations of Mount Kinabalu, Crocker and Trusmadi Range, Schwaner Mountains and Meratus Mountains; (B) Specimens localities of Everetia baramensis, E. jucundior, E. klemmantanica, E. lapidini, E. planispira, E. algaia, E. sp. 1, E. sp. 2, E. sp. 3, E. sp. 4, E. sp. 5, E. sp. 6, E. sp. 7, and Quantula striata; (C) Specimens localities of E. interior, E. jucunda, E. layanglayang, and E. paulbasintali; (D) Specimens localities of E. subconsul, and E. themis.

Figure 2 The distribution of selected Everettia species and specimens of Mount Kinabalu, Sabah for phylogenetic analysis. The numbers in parentheses refer to specimen numbers of Table 1.

(A) Topography of Sabah and location of Mount Kinabalu (red square); (B) Specimens localities of Everetia corrugata corrugata, E. c. corrugata, and E. dominiki; (C) Specimens localities of E. jasilini, E. monticola, and E. safriei.

For species distribution modelling, we obtained distribution records of Everettia species from the BORNEENSIS Molluscan collection that consists of 860 collection lots of Everettia species from Sabah that were collected between the years 2000 and 2018 (Figs. 3–6). After excluding collection lots for which the exact location and species identity could not be determined, the final distribution data consists of 718 collection lots, which comprise 2,024 specimens of 17 Everettia species from Sabah (Additional File 1). The sampling bias in the distribution data from BORNEENSIS collection is negligible as the entire surface of Sabah has been covered adequately in terms of the geographical space, with some areas having been sampled more densely due to the heterogeneity of the habitat such as mountain ranges and islands (Figs. 3–6).

Figure 3 Contemporary distribution records, estimated habitat suitability area of present and Last Glacial Maximum (LGM) bioclimatic conditions for four Everettia species.

(A) Distribution records of E. safriei; (B) Present habitat suitability area for E. safriei; (C) LGM habitat suitability area for E. safriei; (D) Distribution records of E. jasilini; (E) Present habitat suitability area for E. jasilini; (F) LGM habitat suitability area for E. jasilini; (G) Distribution records of E. corrugata williamsi; (H) Present habitat suitability area for E. c. williamsi; (I) LGM habitat suitability area for E. c. williamsi; (J) Distribution records of E. corrugata corrugata; (K) Present habitat suitability area for E. c. corrugata; (L) LGM habitat suitability area for E. c. corrugata.

Figure 4 Contemporary distribution records, estimated habitat suitability area of present and Last Glacial Maximum (LGM) bioclimatic conditions for four Everettia species.

(A) Distribution records of E. layanglayang; (B) Present habitat suitability area for E. layanglayang; (C) LGM habitat suitability area for E. layanglayang; (D) Distribution records of E. dominiki; (E) Present habitat suitability area for E. dominiki; (F) LGM habitat suitability area for E. dominiki; (G) Distribution records of E. lapidini; (H) Present habitat suitability area for E. lapidini; (I) LGM habitat suitability area for E. lapidini; (J) Distribution records of E. monticola; (K) Present habitat suitability area for E. monticola; (L) LGM habitat suitability area for E. monticola.

Figure 5 Contemporary distribution records, estimated habitat suitability area of present and Last Glacial Maximum (LGM) bioclimatic conditions for four Everettia species.

(A) Distribution records of E. paulbasintali; (B) Present habitat suitability area for E. paulbasintali; (C) LGM habitat suitability area for E. paulbasintali; (D) Distribution records of E. occidentalis; (E) Present habitat suitability area for E. occidentalis; (F) LGM habitat suitability area for E. occidentalis; (G) Distribution records of E. jucunda; (H) Present habitat suitability area for E. jucunda; (I) LGM habitat suitability area for E. jucunda; (J) Distribution records of E. interior; (K) Present habitat suitability area for E. interior; (L) LGM habitat suitability area for E. interior.

Figure 6 Contemporary distribution records, estimated habitat suitability area of present and Last Glacial Maximum (LGM) bioclimatic conditions for three Everettia species.

(A) Distribution records of E. jucundior; (B) Present habitat suitability area for E. jucundior; (C) LGM habitat suitability area for E. jucundior; (D) Distribution records of E. planispira; (E) Present habitat suitability area for E. planispira; (F) LGM habitat suitability area for E. planispira; (G) Distribution records of E. subconsul; (H) Present habitat suitability area for E. subconsul; (I) LGM habitat suitability area for E. subconsul.

Molecular methods

Genomic DNA from approximately 2–3 mm3 of foot tissue of single individuals (either fresh, frozen, or kept in ethanol) was extracted with DNeasy™ nucleic acid extraction kits (QIAGEN®, Hilden, Germany) and subsequently stored at −20 °C. Then, PCR was performed using a PTC-200 thermocycler (MJ Research, Inc., St. Bruno, QC, Canada) or T100™ Thermal Cycler (BIO-RAD, Hercules, CA, USA) to amplify the mitochondrial DNA regions 16S with the primer pair 16Sbr-L and 16Sbr-H (Palumbi et al., 1991) and COI with primers LCO1490 and HCO2198 (Folmer et al., 1994). Also, the nuclear rDNA region ITS-1 was amplified with the primer pair 5.8c ‘silkworm’ and 18d ‘fruitfly’ (Hillis & Dixon, 1991) and 28S with primers 28S1128 and 28S2119R (De Weerd, 2008). PCR reactions were performed in 50 μl volumes, using 5 μl 10 × reaction buffer (PROMEGA® or QIAGEN®), 5 μl two mM dNTP, 6 μl 25 mM MgCl2, 2 μl for each primer (5 pmol), 26.85 μl de-ionized autoclaved water and 1 unit of Taq polymerase (PROMEGA® or QIAGEN®). Later, the following cycling profile was used: 2 min at 95 °C, followed by 35 cycles of 1 min at 95 °C, 1 min at 55 °C for 16S, COI and 28S (60 °C for ITS-1) and 2 min at 72 °C, and a final extension period of 10 min at 72 °C. Next, PCR-amplified DNA fragments were purified with the High Pure PCR Product Purification Kit (Roche® or ExoSAP-IT®), according to the manufacturer’s protocol. Finally, DNA sequencing was performed directly on purified PCR products in both directions using the BigDye Terminator Cycle Sequencing Kit v. 3.1 (Applied Biosystems Ltd., Waltham, MA, USA), on an ABI 3100 Genetic Analyser (Applied Biosystems Ltd., Waltham, MA, USA), by Macrogen® or the BigDye® Terminator v1.1, v3.0 and v3.1 Sequencing Kit on an Applied Biosystems 3730xl DNA Analyser at MyTACG Biosciences Enterprise.

Phylogenetic analysis

A total of 96 genetic sequences of the previous study (Liew, Schilthuizen & Vermeulen, 2009) and 160 new genetic sequences from the present study were aligned using the ClustalW multiple alignment algorithm in the BioEdit Sequence Alignment Editor, version 7.0 (Hall, 1999) and manually adjusted with the same programme. Before the phylogenetic analyses, the data matrix was partitioned by markers and codons of COI, namely, first, second and third codon positions of COI, 16S rDNA, ITS-1 and 28S rDNA. Then, each of the partitions was tested for molecular evolution via ModelFinder (Kalyaanamoorthy et al., 2017) and partition models (Chernomor, Von Haeseler & Minh, 2016) based on the both AIC and BIC that built into IQ-Tree v.1.6.7 (Nguyen et al., 2015; Trifinopoulos et al., 2016). We limited the candidate models to the six models that are available in MrBayes analysis, namely, JC, F81, K80, HKY, SYM and GTR. The results of ModelFinder and partition model suggested different partition schemes and substitution models for respective AIC and BIC selection criteria (Additional File 2). We explored the phylogenies estimated based on different substitution models selected for AIC and BIC but the resulted phylogenies are generally congruent (Additional File 3). Hence, we used the best-fit substitution models and partition scheme of BIC selection: partition (1) 16S+ITS: GTR+F+G4, partition (2) COI1+COI2+28S: SYM+I+G4 and partition (3) COI3: GTR+F+G4.

The sequences were analysed using Bayesian analysis (BA) with MrBayes 3.1 (Huelsenbeck & Ronquist, 2001) at the CIPRES Science Gateway portal (Miller, Pfeiffer & Schwartz, 2010) and a maximum likelihood (ML) method implemented in IQ-Tree v.1.6.7 (Nguyen et al., 2015). For BA, the data matrix was analysed with 10 million generations and sampled every 1,000th generation. Then, we discarded the first 25% of the samples. BA was repeated three times for data matrix, and a consensus tree with a cut-off value of 50% was calculated for the resultant trees. For ML analysis, we estimated the phylogeny by using 1,000 ultrafast bootstrap replicates (Minh, Nguyen & von Haeseler, 2013).

Estimation of divergence time

BEAST 2 (ver. 2.6.1) (Drummond & Rambaut, 2007) was used to estimate the timescale for Everettia species divergences based on selected samples for each species. We presume that the split between two Everettia species: E. sp. 1 and E. sp. 2, that occur at the two sides of the Meratus range in South Kalimantan based on a geological event - the uplift of the Meratus Range during late Miocene (10 Ma) (Hall, 2013). Hence, the hypothesis on the timing of speciation of the phylogeny is based on this calibration point which the divergence of the species has resulted from the uplifting of the mountain ranges in Borneo. The tools provided in BEAST 2 were used to estimate node ages to the most common recent ancestor of the split and substitution rates.

We carried out four independent runs of 50,000,000 generations each, sampled every 10,000 generations, using calibrated Birth-Death model with best-fit GTR models, a relaxed lognormal molecular clock was employed, and default options for all other priors and operator settings. The Birth–Death model is chosen as we believe that the evolution of Everettia species a continuous-time process with a probability that a lineage will go extinct. We also explored the time divergence estimates for the combinations two different best-fit substitution models (selected by BIC and AIC criteria) and two calibrated models (Yule model vs Birth–Death model) and the results of these analyses are similar (Additional File 4). The output of each independent run was visualised in Tracer 1.4. Samples and trees from separate runs were pooled after removing the first 10% as burn-in using LogCombiner ver. 2.6.1 and 10% of the trees were discarded as burn-in, and maximum clade credibility trees were calculated each from the remaining 180,004 trees using TreeAnnotator 2.6.0. Divergence dates were computed using BEAST 2 at CIPRESS. The geology-based calibration point (10.0 Ma ± 0.5, 95% CI) was taken as the central trend of a normally distributed prior in BEAUti.

Ecological-niche modelling

To understand how the distribution of Sabah Everettia species has changed during the paleoclimatic fluctuations in the Pleistocene, we predicted ecological niches for all eighteen Sabah Everettia species by using current distribution data under the contemporary (i.e. interglacial) and past (i.e. glacial) climatic conditions. As in other land snail studies (Hugall et al., 2002), we assumed niche conservatism for Everettia.

For the environmental data, we used the bioclimatic dataset version 1.4 (http://www.worldclim.org/current; Fick & Hijmans, 2017). Each of the current bioclimatic layers of resolution of 30 arc-s was clipped to the extent of Borneo. After that, we sampled bioclimatic variables for 500 random locations in Borneo to evaluate the collinearity among the 19 climatic variables by using pairwise Pearson’s r correlation (Additional File 5). After we removed highly correlated variables (r > 0.8), a total of seven climatic variables were used for species distribution modelling, namely, BIO1 Annual Mean Temperature, BIO3 Isothermality, BIO4 Temperature Seasonality, BIO7 Temperature Annual Range, BIO12 Annual Precipitation, BIO15 Precipitation Seasonality and BIO19 Precipitation of Coldest Quarter. Next, the corresponding seven bioclimatic variables of the paleoclimatic dataset for the LGM (model CCSM; http://www.ccsm.ucar.edu/, Kiehl & Gent, 2004) were resampled at resolutions of 30 arc-s (~1 km2).

Then, MaxEnt software (ver. 3.4.1, Phillips, Anderson & Schapire, 2006; Phillips & Dudík, 2008) was used to generate logistic probability maps of species presence with logistic values ranging from 0 (unsuitable) to 1 (optimal habitat). The model was run using the following settings: the maximum number of background points = 10,000; replicates = 10; and replicate run type—Cross validate. All other parameters were kept at default values. Finally, the average of the logistic probability of species occurrence for each grid cell was calculated from the resultant ten replicates.

Results

Phylogenetic analyses

The combined mitochondrial and nuclear DNA matrix comprises 73 specimens and 2,795 characters (16S: 1–501; COI: 502–1059; 28S: 1060–1869; ITS: 1870–2795 (Additional File 6). The best nucleotide substitution models are reported in Additional File 2. As revealed by the Bayesian posterior probability (PP) and maximum likelihood analysis bootstrap (BS) values of the phylogenetic tree in Fig. 7, most of the species are monophyletic, and phylogenetic relationships between species are similar to those found in a previous study (Liew, Schilthuizen & Vermeulen, 2009).

Figure 7 The phylogeny of 25 Everettia species with Quantula striata as outgroup.

Bayesian inference 50% majority-rule consensus trees based on the concatenated dataset consisting of parts of 28S, ITS-1, COI and 16S. Bayesian posterior probabilities and bootstrap support after 1,000 maximum likelihood replicates are shown above and below the branches of the nodes. The font and colour of the taxa name on the tree indicate the distribution of the species. The colour panels next to the taxa names indicated the lowest elevation distributional (Left) and highest elevation distributional (Right) of the species. The number after the taxa name specimen number of Table 1; Figs. 1 and 2.

In contrast to the previous study (Liew, Schilthuizen & Vermeulen, 2009), this study shows the phylogenetic relationship of Sabah Everettia species in the broader context of Bornean Everettia species. Everettia species of Sabah do not form a monophyletic group, and belong to four independent lineages, namely: lineages A, C, D and E (Fig. 7). The other lineage B consists of one species from Sarawak near the border with Brunei and two species from South Kalimantan. However, some of the phylogenetic relationships among these lineages are poorly supported by Bayesian analysis (i.e. PP < 0.95) (Fig. 7).

A total of 12 out of 16 Everettia species in Sabah belong to two major lineages. The first lineage (hereafter, lineage A) consists of nine species, seven of which are lowland species that have their lowest elevation distribution below 1,000 m, namely E. subconsul, E. interior, E. paulbasintali and E. jucundior from Sabah (Figs. 5–7); E. algaia from Sarawak; E. sp. 4 and E. sp. 5 from East Kalimantan (Fig. 1). Two of the species of this lineage (E. layanglayang and E. themis) have their lowest elevational distribution below 2000 m (Figs. 4 and 7). With this expanded genetic dataset, E. themis is now paraphyletic to E. subconsul.

The second lineage (hereafter, lineage C) consists of eight species, of which four are Mount Kinabalu endemics with a lowest elevational limit above 2,000 m, namely E. jasilini, E. safriei, E. corrugata corrugata, and E. c. williamsi (Figs. 3 and 7); two are highland species with their lowest elevation above 1,000 m, namely E. monticola and E. dominiki; and a further two are lowland species: E. planispira from Sabah and E. sp. 3 from Central Kalimantan which occur more than 600 km apart from each other (Figs. 1 and 6).

The remaining four Sabah Everettia species, namely E. jucunda, E. klemmantanica, E. lapidini and E. jucundior, do not belong to the lineages A and C. The Sabah Everettia jucunda form a lineage with an Everettia species (sp. 6) from Sarawak. The Sabah and Sarawak species are more than 500 km apart from each other (lineage E, Fig. 7). E. lapidini and E. klemmantanica are not shown as mutually monophyletic species but as a joint monophyletic clade (lineage D, Fig. 7).

The lineage B consists of two Everettia species from South Kalimantan (sp. 1 and sp. 2) and E. baramensis from Sarawak. The Sarawak and South Kalimantan species are more than 700 km apart from each other (lineage B, Fig. 7). Lastly, E. sp. 7 from Sarawak does not form a clade with any other Everettia species.

Divergence time and tempo of speciation

Here, we used only one calibration point based on a single biogeographic event given the limited availability of fossil records, and reasonable estimates of mutation rates across different genes for the land snail taxa in this region. Currently, the only known land snail fossils in Southeast Asia are from species of Family Cyclophoridae that cannot be used for calibration in this study (Raheem et al., 2018; Xing et al., 2019). The topography of the chronogram is generally congruent with the phylogenetic analysis, of which most of the deeper nodes are poorly supported (PP < 0.95) (Fig. 7; Fig. 8). Our results show the divergences among Everettia in various areas of Borneo are tally to the area’s major mountain uplifting events. These divergence time estimates are based on the hypothesis that mountain uplifting events caused the divergence of the two Everettia species at the two sides of the Meratus range could be falsified in the future if there are more accurate vicariance geological events or reliable fossil record available to improve the calibration of the phylogeny. Diversification of Everettia species in Borneo began in the Late Oligocene (25.8 Ma). These species diversified into five major lineages between the early Miocene (23–17 Ma). The lowland lineage (lineage A) diversified rapidly into seven species between 7 and 19 Ma (Fig. 8). The highland lineage (lineage C) diversified rapidly into montane species and the Mount Kinabalu endemics lineage between 4 and 15 Ma (Fig. 8). Deep divergence of the South Kalimantan and Sarawak species is seen in lineage B (17 Ma).

Figure 8 The chronogram for Everettia species in Borneo obtained from divergence time estimation using BEAST.

The divergence times (in million years ago, Mya) of the major lineages are shown as values on the chronogram branches: bold values are mean ages and values between brackets represent the 95% Highest Posterior Density (HPD) interval (i.e. bar values). The arrow indicates the calibration points. The font and colour of the taxa name on the tree indicate the distribution of the species. The colour panels next to the taxa names indicated the lowest elevation distributional (Left) and highest elevation distributional (Right) of the species. The number after the taxa name specimen number of Table 1, Figs. 1 and 2. Bottom shows the palaeogeography of Borneo: 25 Ma—Late Oligocene. A large part of Borneo was submerged, except the area of West Kalimantan; 20 Ma—Early Miocene. Increase of land area in central Borneo and uplift of the central Borneo mountains; 15 Ma—Middle Miocene. Further uplift in central Borneo and northern Borneo, much of present-day Sabah was below or close to sea level and probably with a minor elevated range of hills at the position of the Crocker range; 10 Ma—Late Miocene. Further uplift of the central part of Borneo, propagation of land area in eastern and northern Borneo with the gradual rise and widening of the Crocker Range, and uplift of Meratus mountains in South Kalimantan. Borneo was now a significantly emergent and elevated area.; 5 Ma–Early Pliocene. Further propagation of land area in eastern, southern and northern Borneo. Image source: Hall, 2013.

Comparison of the ecological-niche model of contemporary and past distribution

Distributional range shifts of Everettia species during the LGM are predicted by the habitat suitability maps in Figs. 3–6. As shown in the phylogenetic analysis, E. themis is now considered as E. subconsul for species distribution modelling (Fig. 7). The area under the curve (AUC) values for the 15 species models are higher than 0.85, except E. klemmantanica.

Most of the Sabah Everettia species have their suitable habitat in Sabah, particularly the endemic species on Mount Kinabalu and central mountain ranges in Sabah. The analysis suggests that suitable habitats for E. jucundior, E. planispira and E. interior are not limited to Sabah, but are extended to large areas in the eastern and southern part of Borneo. Besides, small areas of suitable habitats for E. paulbasintali, and E. occidentalis are located in the eastern part of Borneo.

The palaeoclimatic models predict contraction and expansion of suitable habitats during the LGM for different Everettia species. All four Mount Kinabalu endemic species (Everettia corrugata corrugata, E. c. williamsi, E. jasilini and E. safriei) have experienced range expansion during the LGM at the central mountain range of Sabah. Highland species E. dominiki, E. monticola, E. layanglayang and E. lapidini experienced range expansion as Mount Kinabalu endemics and also in the mountain ranges in the western Borneo.

Everettia planispira, the lowland relative of E. dominiki and E. monticola—experienced significant range expansion in eastern and southern Borneo. Phylogenetic analysis suggests that E. planispira is the sister taxon for E. sp. 3, which is found in southern Borneo. A few of the lowland species, viz. E. paulbasintali, E. occidentalis, E. jucunda, and E. jucundior, experience range contraction and probably remain with very limited suitable habitats. E. subconsul was predicted to have experienced a shrinking of suitable habitat during the LGM into areas near the tip of northern Borneo, including offshore islands and lowland around Mount Kinabalu. Phylogenetic analysis also showed that the populations of E. subconsul on northern offshore islands and tips of northern Borneo are the oldest for the species.

The other lowland species E. interior experienced a little reduction of suitable habitats and its contemporary distribution range is similar to that during the LGM. In particular, the contemporary distributional range of E. interior could potentially extend to eastern Borneo.

Discussion

A high species diversity and high degree of endemism in northern Borneo are well known for many plant and animal taxa, particularly for the central mountain ranges, that is the Crocker Range, Mount Kinabalu and the Trusmadi Range (Liew, Schilthuizen & Vermeulen, 2009; Beaman, 2005). Land snail studies in other regions suggest that vicariance events that persist long enough play crucial roles in driving radiation (Douris et al., 1998; Parmakelis et al., 2005; Fiorentino et al., 2010; Pfenninger et al., 2010; Rowson, Tattersfield & Symondson, 2011), with other factors such as dispersal events and niche differentiation causing further modification (Douris et al., 1998; Schilthuizen et al., 2004; Hausdorf & Hennig, 2004, 2006; Holland & Cowie, 2009; Ketmaier et al., 2010; Kokshoorn et al., 2010). Previously, the phylogeny of Everettia species was estimated without other congener species from outside of Sabah (Liew, Schilthuizen & Vermeulen, 2009). Although species sampling outside of Sabah is still far from complete, these additional species from part of Borneo provides a more accurate phylogeny to illustrate the evolution of Sabah Everettia species that are more or less completely sampled reveal several novel insights.

Divergence of species in the highland lineage

First, most of the Sabah species belong to two deeply diverged lineages. One lineage mainly consists of highland species, particularly all endemics of Mount Kinabalu, while the other lineage includes lowland species. The divergence of these two lineages took place during the early Miocene, which coincided with the uplift of mountain ranges and an extended land area from the southwest to the northeast of the centre of Borneo (Fig. 8). Hence, the divergence was not caused by the more recent uplift of Mount Kinabalu as postulated by studies on other organisms (O’Connell et al., 2018).

The diversification of the four Kinabalu endemics (E. jasilini, E. safriei, E. corrugata and E. c. williamsi) within the highland lineage happened after the middle Pliocene (after 3.8 Ma), and could have been caused by the uplifting of Mount Kinabalu (Figs. 3 and 4). The rapid uplift of Mount Kinabalu at the rate of 500 m per million years (Cottam et al., 2010) could have caused allopatric speciation when the habitat at higher elevation arose, and populations were isolated (Merckx et al., 2015). However, the remaining three species (E. monticola, E. dominiki and E. layanglayang) that reach to an elevation of 3,000 m on Mount Kinabalu and are sympatric with the four Kinabalu endemics more likely diverged by geographical isolation on other mountain summits and subsequently became secondarily sympatric (judged by their deep divergence, before the emergence of Mount Kinabalu).

The palaeo-distributions during the LGM of these seven species provide some insights that these species had more widespread distribution ranges in the central mountain ranges of Borneo that are adjacent to Sabah, based on the suitable habitat analysis (Figs. 3 and 4). This suitable habitat may have facilitated dispersal of these once geographically isolated highland species between Central and northern Borneo montane areas when the cooler temperature during the LGM caused the montane forest to descend and spread, which would have increased connectivity among mountains (Manthey et al., 2017).

However, habitat at lower elevations became hostile to these highland lineage species when the climate warmed up during interglacials. These species probably reacted by moving to suitable habitat at higher altitudes or went extinct altogether. Thus, we believe that Mount Kinabalu has served as a refugium during interglacial periods for highland Everettia species. These highland species could have been trapped there during several glaciation cycles, although we cannot say at which Quaternary glaciation stages this happened. Furthermore, we have shown that land snails on other northern Bornean mountains also show shorter ranges at higher elevations compared to the lowland and lower montane areas (Liew, Schilthuizen & Lakim, 2010), indicating that these species have been pushed upwards until the end of their optimum habitat. This finding supports the studies of other taxa that proposed the mountain ranges in Sabah play a role in the maintainance of ancient lineages (Sheldon, 2017).

The discrepancy of the two divergent processes for the sympatric species on top of Mount Kinabalu provides additional insight that challenge the conventional view that Mount Kinabalu acted as a ‘speciation pump’ and that lower elevation ancestors gave rise to high-elevation endemics (Lee & Lowry, 1980; De Laubenfels, 1988; Holloway, 1996; Chan & Barkman, 1997; Barkman & Simpson, 2001; Tanaka et al., 2001).

Divergence of species in the lowland lineage

Sabah became fully emergent only at the end of the Miocene or Early Pliocene. Two of the most widespread lowland species in Sabah—E. subconsul and E. paulbasitali from lineage A, rapidly colonised newly emerged habitat. Although we did not perform analysis on different populations of other lowland species, we think it is very likely the other widespread lowland species, for example E. jucundior and E. planispira, dispersed to the newly formed land at the same time. In addition to the role of Mount Kinabalu as an interglacial refugium for highland lineage species, SDM analysis shows that Mount Kinabalu also acted as a glacial refugium for lowland lineage species, for example E. subconsul. northern Borneo has been mentioned as a probable glacial refugium during climate changes in the Pleistocene (Brandon-Jones, 1996, 1997; Gathorne-Hardy et al., 2002), but the exact locations of suitable refugia have remained unknown, with some hypothesising that Mount Kinabalu and the Crocker Range could have played such a role (Cockburn, 1978; Smith, 1980; Quek et al., 2007; Jalil et al., 2008). Our study identifies two probable glacial refugia for E. subconsul, on the east and west slopes of Mount Kinabalu. These two glacial refugia, together with unsuitable habitat and mountain ranges as geographical barriers in the centre of northern Borneo, could explain how the east and west coast populations of E. subconsul have maintained their deeply diverged origin since the late Miocene (Figs. 6 and 8).

In contrast to the distribution patterns in the highland lineage, most of the species in lowland lineage occur allopatrically, with the exception of E. subconsul, E. paulbasintali, E. planispira and E. jucundior which are sympatric on the east coast of Sabah. At first glance, the allopatric distributions of the lowland Everettia species appear to be due to geographical isolation caused by mountain ranges, as has been suggested in studies on other taxa (Bänfer et al., 2006). Besides, distribution patterns of lowland species are similar to physiography, vegetation and biozoographical subregions of northern Borneo (Collenette, 1963; Mackinnon & Mackinnon, 1986; Mackinnon et al., 1996; Wong, 1998).

Based on the palaeo-distribution analysis, the lowland species mostly expanded post-glacially, whereas the ranges of the highland species are currently contracting by moving to higher elevations. These different responses by highland and lowland land snails to climate fluctuations are also known from other tropical regions (Wronski & Hausdorf, 2008). Hence, Mount Kinabalu acts as interglacial and glacial refugium for remnant populations, which results in a species diversity hotspot. In Everettia, a total of thirteen out of seventeen northern Borneo species occur on Mount Kinabalu, and six of those species are endemic. The high richness of ancient species agrees with the fact that northern Borneo has had a stable ever-wet climate with most of the forest persisting over the glaciations (Bird, Taylor & Hunt, 2005; Wurster et al., 2010). northern Bornean populations or taxa are known to have been isolated from other parts of Borneo, especially western Borneo, in rainforest refugia during the Pleistocene (Moyle et al., 2005, 2011, 2017; Sheldon, Lim & Moyle, 2015).

Conclusions

Our data enhance the understanding of the evolutionary history of northern Borneo. The northern Borneo Everettia species belong to two deeply diverged lineages. The ecological differentiation and divergence of these two lineages were caused by the uplift of mountain ranges in central Borneo during the Miocene. The continuing eastward and northward extension of Borneo land area together with the formation of central mountain ranges in these newly emerged parts of Borneo have probably driven the species diversification of Everettia in both lineages throughout the Miocene. The species distributional ranges have changed during fluctuating climatic conditions in the Pleistocene. The highland species tended to expand their distribution ranges and lowland species distributional ranges retracted in response to glacial periods, and vice versa during interglacials. We also show that the central mountain ranges of northern Borneo, especially Mount Kinabalu, have acted as refugia in both interglacial and glacial periods. Thus, the contemporary species richness and endemism are caused by geological vicariance events while the contemporary species diversity and distribution patterns are shaped by the Pleistocene climatic fluctuations. We also provide a scenario for how these mountain ranges may have served as refugia for lowland and highland species during both warm and cooler periods. In fact, less than 1% of the total land surface of Borneo is above 2,000 m, and more than three-quarters of this is in northern Borneo. Hence, highland habitats are importance as future refugia for species impacted by rapid climate change in the near future.

Supplemental Information

Supplemental Information 1 Species records of Genus Everettia in Sabah for Maxent Analysis.

The table consists of the 718 records of 17 Everettia species in Sabah obtained from BORNEENSIS collection, Universiti Malaysia Sabah. The collection lot reference number, number of specimens in each collection lot, and the geographic coordinates of the specimens were included in the table.

Click here for additional data file.

Supplemental Information 2 Results of ModelTest for each partition of DNA sequencing alignment and parameter used in BEAST analysis.

Each of the six partitions, namely, codons of COI, namely, 1st, 2nd and 3rd codon positions of COI, 16S rDNA, ITS-1 and 28S rDNA, was tested for molecular evolution via ModelFinder (Kalyaanamoorthy et al., 2017) and partition models (Chernomor, Von Haeseler & Minh, 2016) based on the both AIC and BIC that built into IQ-Tree v.1.6.7 (Nguyen et al., 2015; Trifinopoulos et al., 2016). We limited the candidate models to the six models that are available in MrBayes analysis, namely, JC, F81, K80, HKY, SYM and GTR. Besides, parameters used in BEAST analysis for divergence time estimation were included.

Click here for additional data file.

Supplemental Information 3 Correlations between bioclim variables.

Bioclimatic variables were sampled 500 at random locations in Borneo in each of the 19 climatic layers. Collinearity among the 19 climatic variables was evaluated by using pairwise Pearson’s r correlation. After the analysis, 12 highly correlated variables (r > 0.8) were excluded from MAXENT analysis. The seven climatic variables were used for species distribution modelling, namely, BIO1 Annual Mean Temperature, BIO3 Isothermality, BIO4 Temperature Seasonality, BIO7 Temperature Annual Range, BIO12 Annual Precipitation, BIO15 Precipitation Seasonality, and BIO19 Precipitation of Coldest Quarter.

Click here for additional data file.

Supplemental Information 4 Concatenated DNA Data Matrix for 16S, COI, ITS and 28S Sequences for 73 taxa.

DNA sequences alignment in FASTA format. Position 1–501: 16S; Position 502–1059: COI; Position 1060–1869: ITS; and Position 1870–2795: 28S.

Click here for additional data file.

Supplemental Information 5 Input files and outputs of Bayesian (BA) and Maximum Likelihood (ML) analysis.

The input files and outputs of Bayesian (BA) and Maximum Likelihood (ML) analysis for each of two different best-fit substitution models selected by BIC and AIC criteria, respectively. The phylogenies for each of the analyses were summarised in the word document file.

Click here for additional data file.

Supplemental Information 6 Input files and outputs of BEAST analysis.

A total of four BEAST input XML files for the combinations two different best-fit substitution models (selected by BIC and AIC criteria) and two calibrated models (Yule model vs. Birth-Death model). The calibrated phylogenies for each of the four analyses were summarised in the word document file.

Click here for additional data file.

We thank Jamili Nais, Reuben Clements, and Raes Niels for fruitful discussions. We thank research staff and mountain guides of from the Research and Education Division of Sabah Parks for their field assistance. We are grateful to Maklarin Lakim, Lam Nyee Fan, Noramly Muslim, and Monica Suleiman for providing logistics for field trips and genetic analyses. Wim Maassen (RMNH), Fred Naggs (BMNH), Jonathan Ablett (BMNH), Robert Moolenbeek (ZMA), and Martinah Latim (SP) provided help during our visits to their respective institution collections. We appreciate constructive comments from Nikolay Poyarkov, Frank Koehler and an anonymous reviewer during the peer review.

Additional Information and Declarations

Competing Interests

Author Contributions

Field Study Permissions

DNA Deposition

Data Availability

The authors declare that they have no competing interests. Jaap Jan Vermeulen is employed by JK Art and Science which is a science illustrator company.

Thor-Seng Liew conceived and designed the experiments, performed the experiments, analysed the data, prepared figures and/or tables, authored or reviewed drafts of the paper, and approved the final draft.

Mohammad Effendi Marzuki performed the experiments, authored or reviewed drafts of the paper, and approved the final draft.

Menno Schilthuizen conceived and designed the experiments, authored or reviewed drafts of the paper, and approved the final draft.

Yansen Chen performed the experiments, authored or reviewed drafts of the paper, and approved the final draft.

Jaap J. Vermeulen performed the experiments, authored or reviewed drafts of the paper, and approved the final draft.

Jayasilan Mohd-Azlan conceived and designed the experiments, authored or reviewed drafts of the paper, and approved the final draft.

The following information was supplied relating to field study approvals (i.e., approving body and any reference numbers):

The research was performed under the research permit of Sarawak Forestry: NPW.907.4.4(Jld.14)-31), WL14/2017; and Sabah Parks: TS/PTD/5/4 Jld.54(112).

The following information was supplied regarding the deposition of DNA sequences:

The 16S sequences described here are available at GenBank: FJ160595 to FJ160596, FJ160598 to FJ160599, FJ160606 to FJ160607, FJ160611 to FJ160614, FJ160617 to FJ160619, FJ160621 to FJ160623, FJ160624, FJ160626, FJ160628 to FJ160630, FJ160634 to FJ160640, FJ160642, FJ160644 to FJ160646, JQ180027 to JQ180029, JQ180031, JQ180033 to JQ180035, JQ180038 to JQ180039, JQ180041 to JQ180042, JQ180049, JQ180053 to JQ180055, MN564843 to MN564862.

The COI sequences described here are available at GenBank: FJ160647 to FJ160650, FJ160657 to FJ160658, FJ160660 to FJ160666, FJ160668 to FJ160671, FJ160673, FJ160675 to FJ160677, FJ160681 to FJ160687, FJ160689, FJ160691 to FJ160693, JQ180061 to JQ180063, JQ180065 to JQ180066, JQ180068 to JQ180070, JQ180072 to JQ180075, JQ180082, JQ180085 to JQ180090, MN564863 to MN564882.

The 28S sequences described here are available at GenBank: JQ180153 to JQ180158, JQ180160 to JQ180168, JQ180170 to JQ180177, JQ180179 to JQ180183, JQ180185 to JQ180186, JQ180188 to JQ180190, MN619662 to MN619677.

The ITS sequences described here available at GenBank: FJ160694 to FJ160697, FJ160700 to FJ160701, FJ160704 to FJ160714, FJ160716, FJ160718 to FJ160720, FJ160722 to FJ160732, JQ180095 to JQ180097, JQ180099 to JQ180100, JQ180102 to JQ180103, JQ180105 to JQ180107, JQ180109, JQ180111 to JQ180114, MN596180 to MN596200.

The following information was supplied regarding data availability:

Data is available at Figshare: Liew, Thor Seng (2019): Molecular phylogenetics of the endemic land snail genus Everettia: implications for the evolutionary history of northern Borneo. figshare. Dataset. DOI 10.6084/m9.figshare.10062371.v1.

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
