# Peer review of "Molecular phylogenetics and evolutionary history of the endemic land snail genus Everettia in northern Borneo"

_PeerJ, doi:10.7717/peerj.9416_

## Round 0.1 · original submission · Major Revisions

Thank you very much for this interesting and important contribution. The two reviewers found merit in your work, but as you can see below, both of them propose a number of important issues that have to be addressed before the paper can be accepted to publication in PeerJ. Both reviewers share their concerns that the current version of the divergence time estimation is probably not very realistic for your dataset. First, I encourage you to follow the suggestion by Reviewer 1 who proposed to reanalyze divergence time using the best-fit substitution model and definitely not the strict clock approach. Second, I would ask you to follow the second reviewers' recommendation to moderate some overstated claims in the Discussion, please check the reviews below. Please thoroughly address all questions raised by the reviewers, since the paper will be sent to them for the second round of review if you decide to resubmit.

Reviewer 1 ·

Basic reporting

Some files of the manuscript did not complete uploaded. No Table 2, Table 3, Additional File 1, Additional File 3, and Figure S1 are not available for review.

Table 1. Please provide the meaning of abbreviation for the voucher specimens. Which museum did the specimen deposit?

Figure 7. It would be more easy to read if the annotation of lowest and highest elevation combined in a single phylogeny. Please refer to Figure 5. in https://doi.org/10.1111/2041-210X.12628. There are some free tools could do this job, such as ggtree or iTOL.

Figure 8. The calibration point should be labeled on the phylogeny.

In file S1 (addition file 2 as the author mentioned in the manuscript), no. 30 and 31 labeled as Everettia jasilini. But no. 30 and 31 were labeled as E. safriei in Table 1 and Figure 2, 7, and 8. Please check these again.

Recommend: To upload the BEAST input XML file and resulting files of Bayesian analysis.

Experimental design

Research question is well defined. In general, sampling data and methodologies could answer the research question. The research could be improved if applied more realistic clock and substitution model for divergence time estimation.

Line 152-154: The evolutionary model of each gene partition was evaluated based on AIC. What’s the results when using BIC? I have try to conduct the Partitionfinder implemented in IQ-TREE. Both AIC and BIC were suggested the combination of 1st and 2nd codon position of COI into a single partition scheme. Different best-fit substitution models were selected by AIC and BIC (as shown in the following paragraph).

Best-fit model according to BIC:
16S: GTR+F+R3
COI codon 1+2: TNe+R3
COI codon 3: K3Pu+F+R3
28S: TNe+R3
ITS: GTR+F+R3

Best-fit model according to AIC:
16S: GTR+F+I+G4
COI codon 1+2: TIM+F+R4
COI codon 3: K3Pu+F+R3
28S: TN+F+R2
ITS: GTR+F+R4

Since the partition scheme and substitution model would have impact on the inferred branch length of the phylogeny, it might have greatly effect on the divergence time estimation. I suggest the author to check the differences of estimated divergence time by using AIC- and BIC-evaluated substitution models.

Line 158: The only substitution model available in RAxML is GTR. It is over-parameterized for your data set. I would suggest to use IQ-TREE to compute the maximum likelihood phylogeny.

Line 174: Divergence time estimation using strict clock model is unrealistic for the data set. Since these snails were assumed to diverge with the ancient geological events, it would be more realistic to choose uncorrelated relaxed clock model or random lock clock model to estimate the divergence time. The application of substitution model also have impact on the estimated branch length as mentioned before. GTR model is the best-fit for the data set.

Validity of the findings

The model setting in divergence time estimation should be modified. It might greatly effect the results of estimated time.

Line 219: Lineage C is not a statistical strong supported clade due to the unstable position of Everettia sp. 7 in bootstrap analysis.

Line 250: No results of the estimated mutation rates available for reviewer.

Line 253: I do not agree with the ‘reliable divergence estimates’ because of unrealistic strict clock model and non-best-fit substitution model were applied for the divergence time estimation.

Additional comments

The article is well-written and proves informative background information of geological setting and distribution pattern of Everettia landsnails. This research provides insights into the formation of Borneo biodiversity, especially for the less researched taxa. However, the file of the manuscript did not complete uploaded. No Table 2, Table 3, Additional File 1, Additional File 3, and Figure S1 are not available for review. I suggest to reanalyze the divergence time using best-fit substitution model and more reliable clock model, rather than strict clock model.

·

Basic reporting

Clear and unambiguous, professional English – Yes (with very minor stylistic imperfections that do not impact the clarity of the manuscript)

Introduction and background provide relevant context in clear and understandable fashion
Figures are relevant and in sufficient quality

Editor to check conformity with PeerJ standards, I don’t consider this as part of my duties

Experimental design

Research question is well-defined, relevant and meaningful

Methods used are (largely, see below) appropriate, I did not detect a significant flaw in their execution, but remark that I have no detailed knowledge of distribution modelling so can’t comprehensively evaluate this part of the manuscript.

BEAST analysis. More details need to be provided about the assumptions that went into the model. Which priors were used has an important effect on the output. Effectively, BEAST provides any tree you wish depending on chosen priors and parameters.
I am not a savvy expert myself, but believe that the birth-death model might have been more appropriate than the yule prior (either way please explain why you chose a certain prior or not). Also, a strict clock is a very risky choice since almost never its assumptions are fulfilled by a certain dataset. Again, here more detailed explanation and justification for the chosen method is needed. It might be prudent to run different scenarios, such as priors and relaxed clocks and inspect the influence on the output.

All this is likely to massively inflate the confidence intervals of the result and may result in less clarity about the dating of speciation events.However, as it stands, the molecular clock analysis pretends an unrealistic accuracy. Therefore, all conclusions based on it are also to be taken with a grain of salt.

Validity of the findings

The authors suggest that their findings support the hypothesis that uplift of the central mountains triggered initial lineage differentiation in this group and that speciation mainly occurred during Pleistocene, not later. However, I see circularity in this argument as they have used the uplift of the mountains to calibrate their chronogram. Obviously, by doing this they deprived themselves from assessing the relevance of this event for the timing of phylogenetic events in Everettia.

If the authors are interested to evaluate the relevance of this uplift, they need to find an extrinsic calibration and check for congruence. Even then, coincidence doesn’t necessarily equal causation.

Furthermore, the uplift of mountains is a slow process that took millions of years to complete (and is still on-going, I suppose). Hence, a confidence interval should be applied to the calibration. To simply use a point calibration at 10 Million years ago is an oversimplification of the facts and appears arbitrary to me.
Check Figure 8: It shows existing highlands already at 25-20 MA.

To solve this dilemma, a series of calibration points was helpful to rain in confidence intervals, but I appreciate that this is probably impossible given the lack of fossils and other suitable calibrations.

If no extrinsic calibration are available, that doesn’t mean doom and gloom. The authors can still postulate that the uplift triggered lineage diversification simply because the observed distributions and phylogenetic relationships are suggestive of this.

Under this premise, they can use their calibrations to formulate a hypothesis on the timing of speciation. This hypothesis stands and falls with the correctness of these assumptions. In the end, it is about how to fashion your argument in a logic way.

Additional comments

The claim of the authors that their study provides “implications for the evolutionary history of northern Borneo” is a bit overstated. In fact, what the authors do is using current knowledge of the geological and climatological history of the island to infer the evolutionary history of the group under study. Nothing wrong about this.

The study provides insights into the evolutionary history of Borneo’s biota only in so far as one might argue that what has been shown for the group under stduy (Everettia) might apply also to other groups. But that’s how far it goes. So, potentially I’d suggest the promise a bit down – perhaps just focus on your group as a potential model for low-agility invertebrate biota.

Overall, I consider the overall story as plausible. Its largest weakness is the molecular clock estimation as it probably underestimating or understating the error that comes with the chosen approach. This can not be easily circumvented. This error may not entirely invalidate the findings of your study, but I would argue that you should use a little more caution in fashioning your arguments by accounting for degree of uncertainty in the divergence time estimation.

---

## Round 0.2 · accepted · Accept

Thank you for taking the time to revise and resubmit your manuscript. I have now read through your paper as well as your letter in response to the reviews. As you can see both reviewers are also very much pleased with the work you did to revise your manuscript. I think that you have very carefully and comprehensively addressed all of the concerns raised, and would like to accept your manuscript for publication in PeerJ.

Thank you for all the hard work you have put into this. Your paper makes a strong contribution to the literature and I look forward to seeing it published. I think I shall cite it in my future papers as well!

Congratulations! And stay safe!

Reviewer 1 ·

Basic reporting

Clear statement for the research.

Experimental design

Well experimental design and improved results of divergence dating.

Validity of the findings

Well stated conclusion and improving our knowledge about the origin of land snail diversity of Borneo.

·

Basic reporting

My impression is that the authors have very carefully and comprehensively addressed all issues raised in reviews of the previous manuscript version. I am satisfied with the outcome and find the article to be written in a clear and concise manner.
The methods applied are suitable and the conclusions drawn from these are sound.
I have n reservations for this study to be published in its present form.

Experimental design

ok

Validity of the findings

ok

Additional comments

well written and informative